

# Influence of the Earth's ring current strength on the Størmer's allowed and forbidden regions of charged particles motion

Alexander S. Lavrukhin, Igor I. Alexeev, and Ilya V. Tyutin

M.V. Lomonosov Moscow State University, Skobeltsyn Institute of Nuclear Physics (SINP MSU), 1(2), Leninskie gory, GSP-1, Moscow 119991, Russian Federation

**Correspondence:** Alexander S. Lavrukhin (lavrukhin@physics.msu.ru)

**Abstract.** Størmer's particles' trapping regions for a planet with an intrinsic dipolar magnetic field are considered, taking into account the ring current which arises due to the trapped particles drift for the case of Earth. The influence of the ring current on the particles' trapping regions topology is investigated. It is shown that a critical strength of the ring current exists, under which further expansion of the trapping region is no longer possible. Before reaching this limit, the dipole field, although deformed, retains two separated Størmer regions. After transition of critical magnitude, the trapping region opens up and charged particles, which form the ring current, get the opportunity to leave it (go to infinity or come to the trapping region from infinity), thus decreasing the ring current strength. Numerical calculations have been performed for protons with typical energies of Earth's radiation belt and ring current. For the Earth case, the Dst index for the critical ring current strength is calculated.

## 1 Introduction

One of the first scientists who studied the problem of finding trajectories of charged particles moving from the Sun to Earth was Norwegian physicist Carl Størmer. Since it is rather difficult to solve this problem directly without numerical methods, he used several simplifying assumptions: only a magnetic field of terrestrial origin acts on a charged particle; magnetic and electric fields from other sources, including charged particle fluxes, are not taken into account (Störmer, 1955). In addition, geometry of this problem was reduced to axial symmetry - the magnetic moment of the dipole is directed along the Earth's axis of rotation. As a result of the analysis, Størmer obtained the boundaries of the allowed regions of motion for particles coming to Earth from infinity.

For a complete analytical solution of the problem, one have to find three integrals of motion. In the given problem the law of conservation of the generalized angular momentum of a particle follows from the isotropy of space (the Lagrangian does not change relative to the coordinate system rotations); the law of conservation of energy is the result of the fact that the Lagrangian does not change with respect to the time shift. The third integral of motion cannot be found analytically. Størmer solved this problem by obtaining an expression of a special parameter $\gamma$, which determines the geometry of allowed and forbidden regions of motion for a charged particle; particles with different energies can have the same $\gamma$ (see Chapter 3). Størmer's analysis have its main application in the prediction of the geomagnetic cutoff of cosmic rays. At the same time, the study of the topology of





the charged particles motion regions is an interesting problem, which allows us to understand the processes occurring in the Earth's magnetosphere at the global level.

The effect of the equatorial ring current on the intensity of cosmic rays was considered by Treiman (1953), where an attempt to determine the changes in the intensity of cosmic rays as a function of the ring current strength has been made. But, the question of how the topology of particles' motion in the magnetosphere changes with growth of the current's strength have not been considered. This work is completely analytical without quantitative estimates. Ray (1956) also carried out an analytical analysis of the equatorial ring current influence on latitudinal changes in the intensity of cosmic rays. As a result, for the initial parameters of the cosmic ray spectrum, the ring current strength magnitude was obtained, which gave the observed latitudinal dependence of the vertical intensity of cosmic rays. The radius of the current ring was about 7.5 Earth radii, and the current strength was sufficient to create a magnetic field depression of 100 nT at the geomagnetic equator. Regions of allowed particles motion have been studied in this paper only in connection with geomagnetic cutoff rigidity calculations. The expression for the current's vector potential was written for a filamentary ring current, because if the current ring has a finite cross-section in this problem, this will greatly complicate it without noticeable change in the obtained results.

Also, Størmer's analysis have been expanded for the external homogeneous field case. Lemaire (2003) introduced a homogeneous and stationary field into the Størmer problem; the orientation of this field was parallel and antiparallel to the dipole magnetic moment. As a result, a new expression that takes into account the additional component of the external field was obtained for the Størmer's potential. Lemaire (2003) showed how the inclusion of the northern and southern homogeneous field affects the allowed and forbidden zones, and also that the southern direction enable solar energetic particles and galactic cosmic rays access to the interiors of the geomagnetic field along the magnetic field lines, which connect the dipole with the interplanetary space.

The main sources which forms the magnetospheric magnetic field are the internal field of the planet, interplanetary magnetic field (IMF) and the field of the magnetospheric current systems (Chapman-Ferraro currents at the magnetopause, magnetospheric tail currents and ring current). In the Størmer problem in connection with its specificity (axial symmetry), we will consider only the contributions of the dipole field (the contributions of the quadrupole and other multipoles have been considered in Shebalin (2004), but only axially symmetric components, which are small for the Earth case), the $b_z$ components of external origin (IMF or tail current sheet field) and field of an axially symmetric ring current. This approximation will allow us to estimate the influence of the ring current on the allowed regions of particles motion for the case of the Earth's magnetosphere. Also, we don't take into account the presence of a magnetopause located at a distance of several Earth radii on the dayside magnetosphere.

So far, Størmer's trapping regions have been considered separately from real trapping regions in the Earth's magnetosphere. We consider allowed regions of charged particles' motion in a wide energy range for different ring current strength magnitudes. We show that using the Størmer's analysis one can find trapping regions for particles of the ring current and radiation belts energies, and also the fact that the forbidden region of particles' motion can break as the current increases, and then particles get an opportunity to leave the trapping region.



The present paper consists of the following chapters: Chapter 2 describes the Earth's ring current and its general features; in Chapter 3 we consider the mathematical formulation of the Størmer problem; in Chapter 4 we consider the effect of the electric field on change of the Størmer's parameter $\gamma$; in Chapter 5 the trapping regions are modeled with different magnitudes of the ring current strength.

## 2   The Earth's ring current

The dipole configuration of the Earth's magnetic field creates a region that is a kind of a "magnetic vessel" inside which charged particles are trapped. For the first time the possibility of the existence of such a region was shown by Carl Størmer (Størmer, 1912). In accordance with the energy, the trapped particles can be divided into three groups: the radiation belt, the ring current and the plasmasphere. The trapping regions in which the particles of each group are contained partially overlap. The energy of the radiation belt ions is between 1 - 100 MeV, they occupy the shells with $L \approx 1.2 - 2.5$ (Prölss, 2004). The energy of the ring current ions is between 1 - 200 keV (Prölss, 2004); it occupies an area with $L \approx 2.3 - 7.8$ (Kovtyukh and Panasyuk, 2008).

For the first time the hypothesis of a ring current was made by Carl Størmer to explain the location of the aurora and its' shift to the equatorial latitudes under magnetic storm conditions (Størmer, 1912). He considered the influence of the ring current located in the Earth's equatorial plane on the structure of the magnetic field and the trajectories of electrons. Størmer showed that if the ring current could create a disturbance of 300 nT on the equatorial surface of the Earth, then the aurora would move to the magnetic latitude of Christiania (district in Copenhagen) (Størmer, 1912).

Magnetic storms on Earth are directly related to the ring current. The increase of its strength leads to compression of the inner magnetic field of the planet. During quiet times, the ring current is localized near the geomagnetic equator, directed to the west and depends little on local time. A geomagnetic storm usually begins with the impact of a strong and permanent south IMF along with the increased dynamic pressure of the solar wind on the dayside magnetosphere. The increased pressure moves the magnetopause inward by several Earth's radii, increasing currents on the magnetopause and temporarily causing an increase of the surface magnetic field. This phenomena is known as the sudden commencement of a storm. During the main phase of the magnetic storm, which is usually associated with the southern IMF magnitude growth, the electric field applied to the magnetosphere increases, which leads to increase in the number of particles injected into the inner magnetosphere due to the $\boldsymbol{E} \times \boldsymbol{B}$ drift from the magnetotail. These particles significantly increase the quiet ring current, eventually creating a so-called storm-time ring current. This leads to a strong Dst index increase and marks the main phase of the storm. The main phase generally lasts for several hours, but for specific storms its duration can vary greatly. After the IMF is rotated to the north, the recovery phase begins, during which the injection of the plasma layer material slows down or almost stops, and the various loss processes return the ring current to its quite state. The dominant source for the ring current is plasma sheet, so the two most important driving parameters for the stormtime ring current are the near-Earth plasma sheet ion characteristics and the convection electric field intensity (Kozyra and Liemohn, 2003).

The strength of the storm-time current is estimated by the Dst index and can reach 10 MA (Baumjohann et al., 2010). For comparison, the magnitude of the region 1 longitudinal currents, which transfer the energy of the solar wind plasma to the

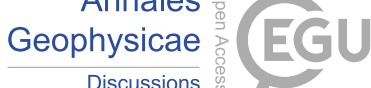



magnetosphere under quite conditions, is 1-3 MA, and the currents at the magnetopause are approximately 2 times larger: $2-6$ MA (Alexeev, 2003). A distinctive feature of the storm-time ring current in comparison with the quiet-time ring current is its pronounced asymmetry. On the nightside, the strength of the storm-time current is much larger than on the dayside. Radial flow and pressure profiles of ring current particles during quiet periods have a maximum at $L_{max} = 5.5 - 6.0$, and during

storms at $L_{max} = 2 - 5$ (Kovtyukh and Panasyuk, 2008). In quiet times, the current mainly consists of protons (average partial concentration $\sim 90\%$, (Daglis et al., 1999). During the magnetic storms there is a sharp increase in the density of atomic oxygen ions $O^+$, originating from the ionosphere. However, protons continue to make the main contribution to the ring current during typical storms, with an average partial energy density of $\sim 70\%$ (Daglis et al., 1999). During very large storms, the $O^+$ ion component becomes predominant (Daglis et al., 1999).

During the main phase of the storm, the hot plasma is injected into the inner magnetosphere. The stronger the storm, the closer the ring current comes to Earth and the slower it relaxes. Storm-time ring current essentially differs from radiation belts, consisting of more energetic particles, in composition, dynamics and generation mechanisms. Unlike radiation belt ions, whose fluxes are weakened during storms (Kovtyukh and Panasyuk, 2008), the fluxes of particles of the ring current increase by one to three orders of magnitude during the main phase of the storms, and after the recovery phase they decrease to some

quasistationary levels.

The ring current is not the only source which contribute to the Dst index during the storm. Alexeev et al. (1996) and Turner et al. (2000) showed that the fields created by the magnetotail currents during the main phase of the storm contribute 25% of Dst, the amount, which, however, significantly decreases during the recovery phase. The magnetic effects of the magnetotail together with the asymmetric ring current give the largest contribution during the main phase (Alexeev et al., 1996; Turner

et al., 2000).

There are several loss processes, which act on decay of the ring current ions: the charge exchange of ions with the exospheric atoms, the Coulomb drag, the Coulomb scattering, pitch angle scattering due to interaction with electromagnetic waves, the violation of the first adiabatic invariant and the convection outflow. All these sources are considered in detail in Ebihara and Ejiri (2003). The outflow of ions through the dayside magnetopause can be a significant term in the energy balance of the

stormtime ring current during the main phase of the storm (Kozyra and Liemohn, 2003). Also, the initial fast stage of the recovery phase of very strong storms can be associated with the decay of the oxygen component, and the subsequent slow stage — with the decay of the proton component (Daglis et al., 1999). Rapid decay of the inner part of the magnetospheric tail current (Alexeev et al., 1996) and the sudden rotation of the IMF to the north at the end of the main phase of the storm were also considered as mechanisms which acts during the fast phase of the recovery of powerful storms, which can lead to an

inversion of the convection electric field and ejection of the ring current from the outer regions of the trapping region (Ebihara and Ejiri, 1998).

We propose a new loss mechanism associated with the disappearance of the Størmer's forbidden region of particles motion between two allowed regions due to an increase in the ring current strength. This consideration allows us to find the upper limit on the maximum current, which can be created by trapped particles. In this threshold case, the ring current particles are allowed

to leave the trapping region, which leads to a weakening of the ring current and Dst index decrease. The existing mechanisms





of scattering and precipitation of particles can lead to the fact that the equilibrium value of the ring current will be smaller (in the case of Earth by more than an order of magnitude) of this threshold value. But can this mechanism limit the maximum field depression during large magnetic storms?

In the zeroth approximation, the ring current can be represented as a ring with current in the equatorial plane at a certain

distance from the Earth. The magnetic field in the center of the ring with a current of radius $R$ is given by (Landau et al., 1982):

$$B = \frac{\mu_0 I}{2R} \tag{1}$$

From this expression, we can obtain an approximation of the magnetic field perturbation on the Earth's surface:

$$\Delta B_{rc} = -\frac{\mu_0 \Delta I_{rc}}{2L R_E} \tag{2}$$

where $r$ is the geocentric distance and $\Delta I_{rc}$ is an increase of the ring current strength. This description is only approximate,

in reality the ring current has a much more complex structure. However, this approach can be used for understanding the processes which occur at changes of the ring current strength and for estimation of the maximum number of particles which may be trapped by the dipole field. Based on the data given in this chapter, we will consider the distance to the ring current from the Earth's center equal to $a = 4R_E$. In connection with the axial symmetry of the Størmer problem, only symmetric ring current in the form of infinitely thin ring will be used.

## 3   The mathematical formulation of the Størmer problem

Equations of the charged particle motion in an electromagnetic field in the case of axial symmetry are more convenient to consider in cylindrical coordinates. We consider a cylindrical coordinate system in which the positive $z$ axis is directed to the north of the planet. The dipole moment $M$ is located at the origin of coordinates and is directed along the negative direction of the $z$ axis, the external magnetic field has only the $z$ component. The radius of the planet is denoted as $R_E$. The radius of

the ring with current is denoted as $a$, the ring is located in the magnetic equatorial plane perpendicular to the $z$ axis, while the origin of coordinates lies in the plane of the ring in its center. In this case, we can speak about the axial symmetry of the magnetic field configuration, since the fields of the dipole and ring current and external field have axial symmetry in this system. We also use the following assumptions used by Carl Størmer: the magnetic field is stationary; we neglect the particle radiation as it moves in a magnetic field.

We will use the Lagrangian formulation, the equation of motion can then be written as (Landau and Lifshitz, 1976):

$$\frac{d}{dt}\frac{\partial L}{\partial \dot{q}_i} - \frac{\partial L}{\partial q_i} = 0$$

The Lagrange function of a charged particle in the relativistic case is (Morozov and Soloviyov, 1963):



$$L = -mc^2\sqrt{1 - \dot{r}^2/c^2} + \frac{Ze}{c}\left(\dot{\boldsymbol{r}} \cdot \boldsymbol{A}(\boldsymbol{r}, t)\right) - e\Phi(r, t)$$

where $\dot{r} = (\dot{\rho}, \rho\dot{\phi}, \dot{z})$ - particle's velocity, $\boldsymbol{A}(\boldsymbol{r}, t)$ and $\Phi(r, t)$ are vector and scalar potentials of an electromagnetic field, $Z$ is the charge of a particle in units of an electron charge.

The scalar potential of an electromagnetic field will not be taken into account due to the absence of an electric field in Størmer's problem. Since the field we are considering has the axial symmetry, in a cylindrical coordinate system the Lagrangian does not depend on the azimuth angle $\phi$, and hence, by the Noether's theorem, the generalized angular momentum (the first integral of motion) is conserved along the trajectory:

$$P_\phi = \frac{\partial L}{\partial \dot{\phi}} = \frac{m\rho^2\dot{\phi}}{\sqrt{1 - \dot{r}^2/c^2}} + \frac{Ze}{c}\rho A_\phi = const \tag{3}$$

The total energy of a particle in the Lagrangian formulation is:

$$H = \dot{r}\frac{\partial L}{\partial \dot{r}} - L = \frac{mc^2}{\sqrt{1 - \dot{r}^2/c^2}} \tag{4}$$

Since we neglect the particle radiation, the total energy $H$ in the constant magnetic field is conserved, and therefore, from (4) we get:

$$\dot{r}^2 = v^2 = const \tag{5}$$

In other words, the square of the velocity of the particle is also a constant in the absence of the electric field (the second integral of motion). We rewrite (5) using velocity components in cylindrical coordinates:

$$\left(\frac{d\rho}{dt}\right)^2 + \left(\rho\frac{d\phi}{dt}\right)^2 + \left(\frac{dz}{dt}\right)^2 = v^2 \tag{6}$$

It is convenient to integrate over the arc of the trajectory $s$ $(ds = vdt)$ instead of integrating over time $t$, and then to divide (6) by $v^2$:

$$\left(\frac{d\rho}{ds}\right)^2 + \left(\rho\frac{d\phi}{ds}\right)^2 + \left(\frac{dz}{ds}\right)^2 = 1 \tag{7}$$

We express $\dot{\phi}$ from (3) and substitute the square root from (4):

$$\rho\dot{\phi} = \frac{c^2}{H}\left(\frac{P_{phi}}{\rho} - \frac{ZeA_\phi}{c}\right)$$



From the relativistic relations, we can obtain the relation $pc^2 = vH$; then we express $c^2/H = v/p$ and proceed from integration over $dt$ to integration over $ds$; finally we obtain:

$$\rho \frac{d\phi}{ds} = \frac{v_\phi}{v} = \frac{P_\phi}{p\rho} - \frac{ZeA_\phi}{pc} \tag{8}$$

After substituting this relation in (7):

$$\left(\frac{d\rho}{ds}\right)^2 + \left(\frac{dz}{ds}\right)^2 + \left(\frac{P_\phi}{p\rho} - \frac{ZeA_\phi}{pc}\right)^2 = 1 \tag{9}$$

For clarity, we rewrite this equation using expression from (8):

$$\left(\frac{d\rho}{ds}\right)^2 + \left(\frac{dz}{ds}\right)^2 = Q = 1 - \left(\frac{P_\phi}{p\rho} - \frac{ZeA_\phi}{pc}\right)^2 = 1 - \left(\frac{v_\phi}{v}\right)^2 \tag{10}$$

where $Q$ is the kinetic energy of particle motion in the meridional plane (Störmer, 1955).

As can be seen from equation (10), during the motion of the particle, the condition $0 \leq (v_\phi/v)^2 \leq 1$ must be satisfied, so that the motion of a charged particle in axially symmetric magnetic field takes place in regions of the meridian plane for which $-1 \leq v_\phi/v \leq 1$ - these are the allowed regions of the particle's motion. The regions for which $(v_\phi/v)^2 > 1$, are forbidden. The boundary between these regions is determined by the equality of the velocity component $v_\phi$ and the velocity of the particle $v$; at this moment particle turns around on the boundary and starts to move in the opposite direction. The existence of the boundaries of charged particles motion regions allows one to make a qualitative analysis of their motion for the chosen $A_\phi(\rho, z)$. In terms of energy, the minimum value of the fraction of kinetic energy $Q$, associated with motion in the meridional plane $(\rho - z)$ is equal to 0; at this moment the velocity of the particle is perpendicular to the meridian plane, and the projection of the velocity on the $(\rho - z)$ plane changes sign, i.e. the particle after reaching the point, where $Q = 0$, turns back. Therefore, the line $Q = 0$ in the $(\rho - z)$ plane limits the region, inaccessible to particles.

Under these conditions, trapping regions, separated from infinity by forbidden regions (where particles can not get due to the preservation of exact integrals of motion for any initial conditions) are formed in the phase space. Depending on the initial conditions, the trajectories of particles of a given energy may have scattering pattern - particles come from infinity, change their direction of motion near the dipole, and again go to infinity, deviating by the scattering angle. Or, if their position and the generalized angular momentum satisfy the Størmer's conditions, the particles make finite motion, filling the allowed Størmer regions and forming the trapped particles fluxes. This moment is given little attention in Størmer's original work (Størmer, 1955), who was mainly interested in the calculation of the penetration regions of cosmic ray particles, i.e. only those particles that come from infinity.

The surface $Q = 0$, which separates the allowed and forbidden regions of motion, is determined from (10):

$$Q = 1 - \left(\frac{P_\phi}{p\rho} - \frac{ZeA_\phi}{pc}\right)^2 \tag{11}$$





The vector potential of the dipole magnetic field is given by:

$$A_{\phi d} = \frac{M\rho}{r^3} \tag{12}$$

The vector potential of a homogeneous magnetic field, directed along the $z$ axis (in our case this field characterize the $z$ component of external field, which may be either IMF, or magnetotail currents' field) is given by (Lemaire, 2003):

$$A_{\phi e} = \frac{b_z \rho}{2} \tag{13}$$

The vector potential of the ring current has a complex form and depends on the assumed radius of its cross section. In our problem we assume that the ring current has the form of a ring with radius $a$ lying in the equatorial plane with the center at the origin, the cross section of which is much smaller than $a$. In this case:

$$A_{\phi r} = -\frac{\pi I a}{c} \sum_{n=0}^{\infty} \frac{(-1)^n (2n-1)!!}{2^n (n+1)!} \frac{r_<^{2n+1}}{r_>^{2n+2}} P_{2n+1}^1 (\rho/r) \tag{14}$$

where $(2n-1)! = (2n-1)(2n-3)\ldots 5\cdot 3\cdot 1$, and the coefficient for the term with n = 0 is by defenition equals 1; $r_<$ and $r_>$ are respectively the smaller and larger of $a$ and $r$ (Jackson, 1963).

One can separate the considered region into two parts: when $\rho < a$ and when $\rho > a$. Then the vector potential (14) will be:

$$A_{\phi r} = -\frac{\pi I}{c} \sum_{n=0}^{\infty} \frac{(-1)^n (2n-1)!!}{2^n (n+1)!} P_{2n+1}^1 (\rho/r) \begin{cases} (\rho/a)^{2n+1}, & \rho < a \\ (a/\rho)^{2n+2}, & \rho > a \end{cases} \tag{15}$$

Substituting all expressions for $A_\phi$ (12), (13), (15) of the vector potential in equation 11, we obtain:

$$1 - \left( \frac{P_\phi}{p\rho} - \frac{Ze}{pc} \left( \frac{M\rho}{r^3} - \frac{\pi I}{c} \sum_{n=0}^{\infty} \frac{(-1)^n (2n-1)!!}{2^n (n+1)!} \cdot \begin{cases} (\rho/a)^{2n+1}, & \rho < a \\ (\rho/a)^{2n+1}, & \rho > a \end{cases} P_{2n+1}^1 (\rho/r) \right) \right)^2 = 0, \tag{16}$$

We define the Størmer radius $r_s$ (Størmer, 1955), the Størmer parameter $\gamma$, and the dimensionless variable $R$, as:

$$r_s = \sqrt{\frac{ZeM}{pc}}; \gamma = \frac{P_\phi}{2pr_s}; R = \frac{r}{r_s} \tag{17}$$

Essentially, $\gamma$ is the ratio of the azimuthal components of two different momenta of the particle: $P_\phi$, taken at infinity, and $pr_s$ at one Størmer radius, with a multiplying factor of 1/2 (taken for the convenience of demonstration); particles of any energy

can have the same $\gamma$. Thus, having two integrals of motion and the Størmer parameter $\gamma$, one can not completely describe the





trajectory of any particle, but can find the regions of allowed motion and, thus, the trapping regions. We rewrite (11) taking into account the newly introduced quantities:

$$Q = 1 - \left( \frac{2\gamma}{\rho} - \frac{ZeA_\phi}{pc} \right)^2 \qquad (18)$$

The size of the allowed regions of motion, as can be seen from (18), will be different for particles with different energies and
with different $\gamma$. The transition to the Størmer units of length allows one to analyze the motion of particles of different energies uniformly. With an increase of the particle energy, its Størmer radius $r_s$ decreases.

Using (17), we can rewrite (16) in dimensionless form, where the units of length are expressed in Størmer units of length:

$$1 - \left( \frac{2\gamma}{\rho} - \frac{\rho}{r^3} - \frac{b_0\rho}{2M}r_s^3 + \frac{\pi I}{Mc}r_s^2 \sum_{n=0}^{\infty} \frac{(-1)^n(2n-1)!!}{2^n(n+1)!} \cdot \begin{Bmatrix} (\rho/a)^{2n+1}, & \rho < a \\ (\rho/a)^{2n+1}, & \rho > a \end{Bmatrix} P_{2n+1}^1(\rho/r) \right)^2 = 0, \qquad (19)$$

After solving (19), one can find the boundaries of the Størmer's allowed and forbidden regions of particles' motion for the
general case of the sum of dipole field, external field and ring current field.

## 4   Influence of the electric field on particle motion in Størmer's theory: particle injection in the inner magnetosphere

The solar wind flow past the magnetosphere generates a large-scale long-standing convection electric field in it, directed from dawn to dusk under the southern IMF direction. As a result, the $\boldsymbol{E} \times \boldsymbol{B}$ drift occurs in the magnetosphere, with the particles moving from magnetotail towards Earth, thus supplying the radiation belts and ring current with particles. Consequently, two
regions appears — in the first, where the gradient drift predominates, there is a constant ring current; in the intermediate region between the gradient and $\boldsymbol{E} \times \boldsymbol{B}$ drift regions, a partial ring current arises due to the fact that protons on one side of Earth, and electrons on another have oppositely directed gradient drift velocities. Thus the partial ring current is most intense on the nightside of Earth. In our problem we consider only symmetric ring current due to the axial symmetry. In addition to the large-scale potential electric fields, that appear in the nightside magnetosphere due to the prolonged southern IMF, the
impulsive induced electric fields (which also can transport particles into the inner magnetosphere) can arise due to the magnetic field reconfigurations of during substorms (Daglis, 2001). These two particle's transport and acceleration mechanisms can act simultaneously Ebihara and Ejiri (2003).

Despite the fact that there are no electric fields in the Størmer analysis, the transport of particles into the inner parts of the magnetosphere can be explained using the change of the Størmer parameter $\gamma$. From (8), (17) we have:

$$\frac{v_\phi}{v} = \frac{2\gamma}{\rho} - \frac{ZeA_\phi}{pc} \qquad (20)$$





The two terms on the right side of (20) represent the effect of centrifugal force and magnetic field respectively, which hold the particle away from the axis. We are interested in particles that can come closer to Earth to form the ring current and radiation belts. In the case of a dipole field, (20) in Størmer units can be rewritten as:

$$\frac{v_\phi}{v} = \frac{2\gamma}{\rho} - \frac{ZeM\rho}{pcr^3} = \frac{2\gamma}{\rho} - \frac{\rho}{r^3} \tag{21}$$

5    If we assume that the particle is injected in the equatorial plane ($z = 0$), then equation (21) can be rewritten as:

$$\frac{v_\phi}{v} = \frac{2\gamma}{\rho} - \frac{1}{\rho^2} \tag{22}$$

Let us analyze this equation. The closer the particle is to Earth ($\rho$ decreases), the larger the second term (magnetic field) on the right-hand side, and, accordingly, the larger the first term must be to compensate the second. Thus, as the particle approaches the Earth, with decreasing $\rho$, its parameter $\gamma$ must increase.

10    In the original Størmer analysis magnetic field is stationary and a particle does not receive energy from outside, so the parameter $\gamma$ will remain constant. In real magnetosphere, the described above $\boldsymbol{E} \times \boldsymbol{B}$ drift exists. Under this drift, a particle moves toward the planet. Electric field is not included in Størmer's analysis, but we can take into account the drift of the particles through the change of $\gamma$, which value will increase when approaching the planet due to the increase of the magnetic field strength. This situation is shown in Figure 1, where the allowed motion regions for the proton with different parameters $\gamma$

15    are shown. The larger $\gamma$, the closer the allowed region of motion shifts to Earth.

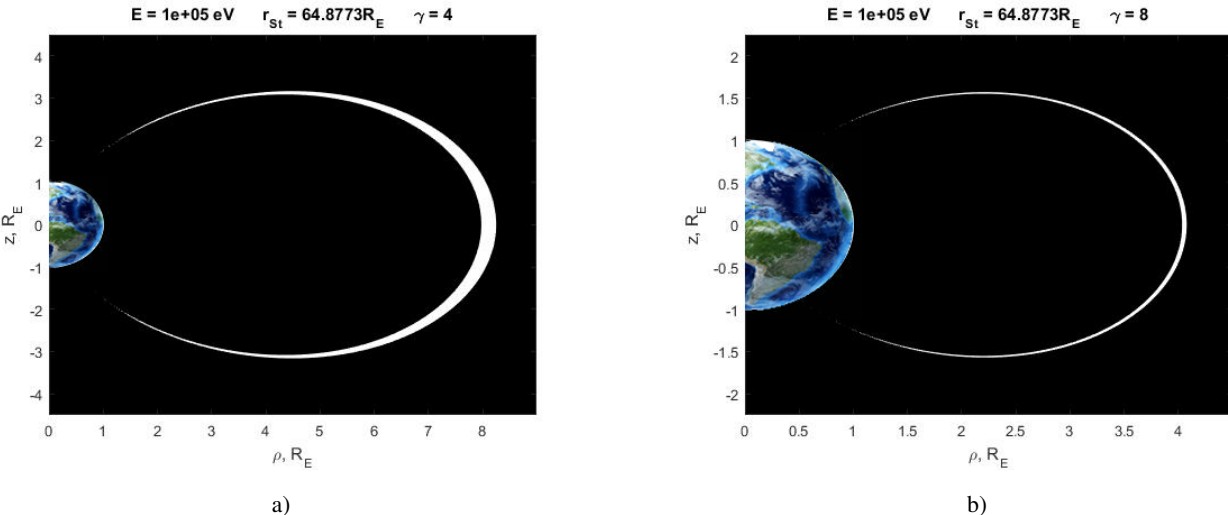

a)    b)

**Figure 1.** Allowed regions of motion of a 100 keV proton for two cases: a) $\gamma = 4$, b) $\gamma = 8$. It can be seen that for $\gamma = 4$, the allowed region of particle motion crosses the equatorial plane at a distance of $8R_E$, and for $\gamma = 8$ - at $4R_E$. Thus with increasing $\gamma$, the particle comes closer to Earth. Allowed region of motion is highlighted in white.





To find $\gamma$ corresponding to a particle with a certain energy located on a certain $L$ shell, it is necessary to use equation (18). It defines the internal and external boundaries of the allowed region of motion; in order to find $\gamma$ for a particle located in the middle of this region, it is necessary to set the terms in brackets to zero:

$$\frac{2\gamma}{\rho} - \frac{ZeA_\phi}{pc} = 0 \tag{23}$$

The equality to $\pm 1$ in equation 23 correspond to the boundaries of the trapping region.

## 5 Modeling of the Størmer's allowed and forbidden regions taking into account the ring current

In modeling, we assumed that:

1. The Earth's magnetic field is constant and has only a dipole component due to the axial symmetry of the problem. This is quite reasonable, because: 1) the dipole magnetic field is responsible for the basic motion of the ring current particles, that is, the bounce and drift motion and 2) the higher degree of the coefficient becomes dominant at radial distance less than $1.5R_E$ or above the South Atlantic and South African areas.

2. The ring current has the form of an infinitely thin ring in the equatorial plane at a distance of $4R_E$ from the center of Earth. This value was chosen in accordance with the spatial dimensions of the ring current in equatorial plane and with average values of the maximum of the ring current density (Kovtyukh and Panasyuk, 2008; Prölss, 2004). We assume that not all of the ring current particles are concentrated in the filamentary ring, but that the magnetic field, created by the real ring current is the same as created by filamentary ring with current.

3. The $b_z$ component of the homogeneous external field is $-15$ nT and describes the field of distant sources: the magnetopause current's field (positive $20 - 100$ nT), the field of tail currents (negative $10 - 200$ nT) and the penetrating interplanetary field (about $1 - 20$ nT of an arbitrary sign).

4. Charged particles both in radiation belt and in the ring current are protons ($H^+$).

5. Radiation belt particles with energies of 1 and 100 MeV.

6. Ring current particles with energies of 10 and 100 keV.

For particles with different values of $\gamma$ and current strengths' $I$, different patterns of the regions' boundaries are obtained. We know that in reality the particles of the ring current are distributed up to $\sim 8R_E$ (see Chapter 2). Using (23), we choose those values of $\gamma$, that describe particles of different energies at a distance $L = 8$. This distance will be considered as the outer edge of the ring current.

All the following figures show the meridional cross sections of the near-Earth region, the allowed regions of motion are highlighted in white, forbidden - in black. Magnetic field lines are highlighted in cyan. Coordinates are measured in the Earth




radii. The spatial configuration of forbidden regions is obtained by rotating the meridional sections around the $z$ axis. Particles cannot cross the forbidden region and, consequently, free particles cannot penetrate into the internal allowed region. Likewise, particles that somehow find themselves in an internal allowed region in the ideal case are permanently trapped.

First we consider the energies of the ring current protons of 10 keV and 100 keV:

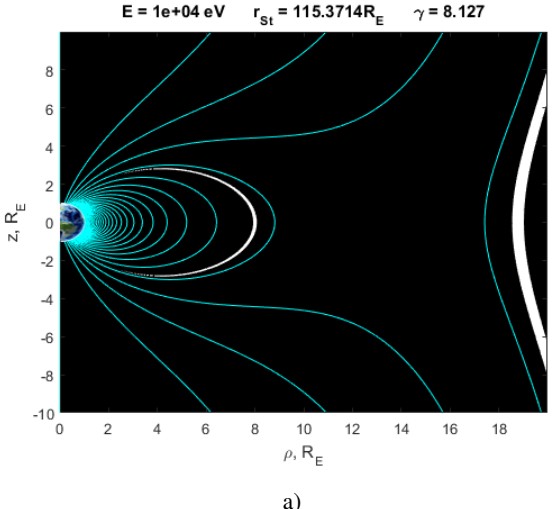

**Figure 2.** The ring current strength $I$ is 0 MA, a) 10 keV, b) 100 keV

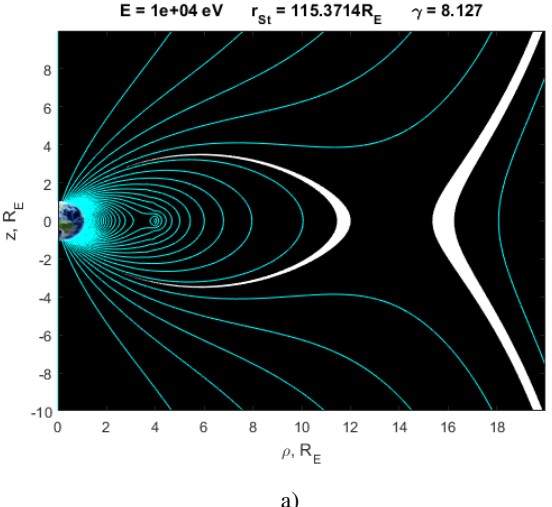
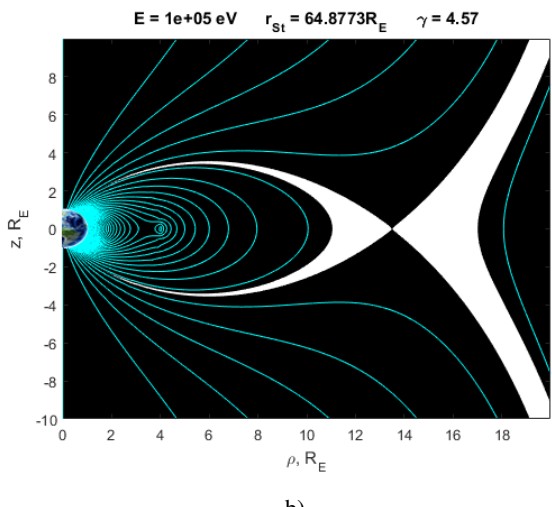

**Figure 3.** The ring current strength $I$ is 9.17 MA, a) 10 keV, b) 100 keV



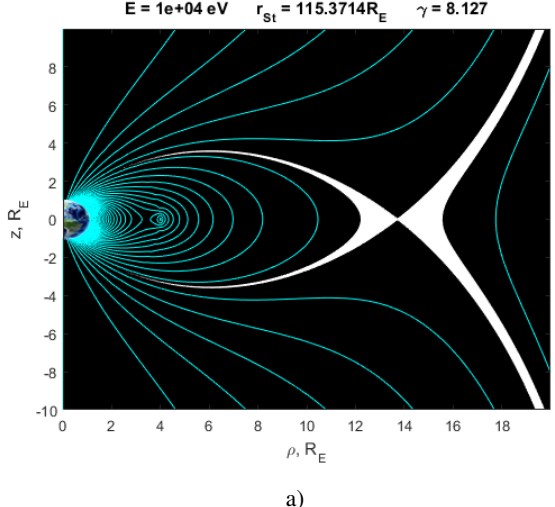
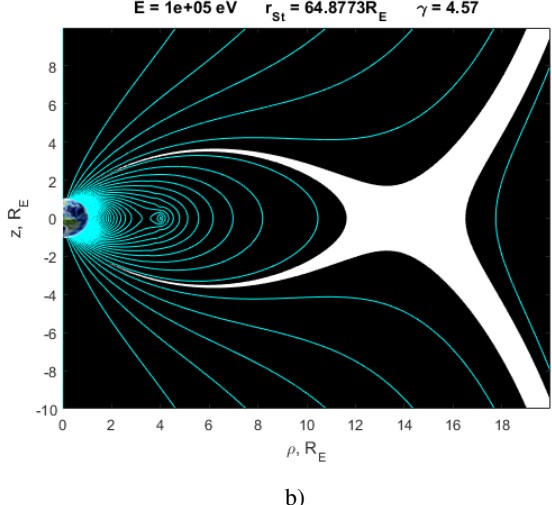

a)                                                    b)

**Figure 4.** The ring current strength $I$ is 10.28 MA, a) 10 keV, b) 100 keV

As can be seen from figures 2-4, with increasing ring current strength, two initially separated (at $I = 0$ A) allowed regions of motion become connected. Moreover, for particles with energies of 100 keV, the escape path from the inner trapping region emerges earlier than for particles with an energy of 10 keV, at a current strength of 9.17 MA (Figure 3b). When current reaches 10.28 MA, particles with energy of 10 keV also get the opportunity to leave the trapping region (4a). As a result, particles of

the outer edge of the ring current with different energies are able to leave the trapping region area.

The obtained current strengths' are consistent with the observations (see chapter 2), and the obtained Dst index values for these current strength magnitudes are 226 and 253 nT, which are not unique and usually characterize large storms. However, we must remember, that Dst index was calculated by equation 2 for an infinitely thin ring with current, i.e. it was assumed that all particles are located in this ring. In reality the situation is different, the particles are distributed in the toroidal volume around

the Earth over a large range of distances; in addition, for the ring current and homogeneous magnetic field the condition of axial symmetry is not satisfied, so the calculation of the real Dst index is much more complicated. Nevertheless, our analysis makes sense, since it gives an upper limit on the maximum ring current strength, that retains the trapping regions. Taking into account the asymmetry will lower this threshold, since the particles will have an opportunity in the nightside to leave the trapping region, going to infinity through the geomagnetic tail, or leaving the magnetosphere through the daytime magnetopause. In

this case, the "bridge" between the trapping region and infinity opens the possibility of injection of particles from the plasma layer into the ring current region with simultaneous acceleration of the particles and an increase of the trapped particles number. However, the obtained magnitudes may probably indicate that the other loss mechanisms begin to act earlier than the mechanism, suggested in this paper. The balance of sources and losses describes the observed dynamics of the magnetic storm.

Now let us consider what happens to the protons of the radiation belts (with energies of 1 MeV and 100 MeV) after increase

of the ring current magnitude (Figures 5-7) for the same magnitudes of the currents chosen for Figures 2-4. The most distant protons of the radiation belt are located at a distance of $\sim 2.5 R_E$ (see Chapter 2).





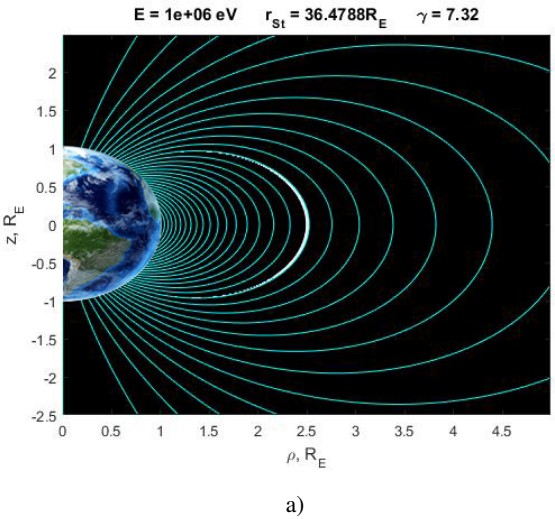

**Figure 5.** The ring current strength $I$ is 0 MA, a) 1 MeV, b) 100 MeV

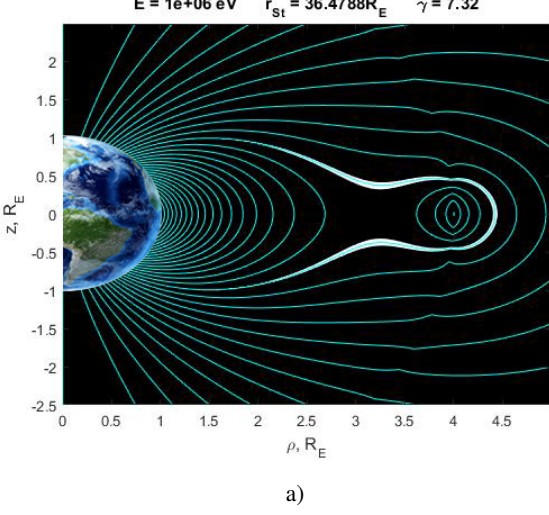

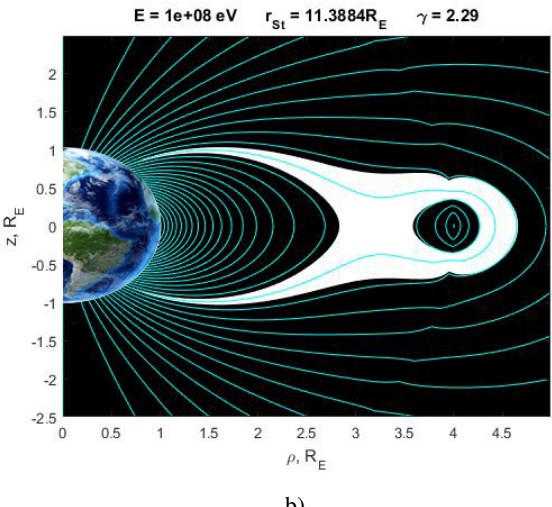

**Figure 6.** The ring current strength $I$ is 9.17 MA, a) 1 MeV, b) 100 MeV



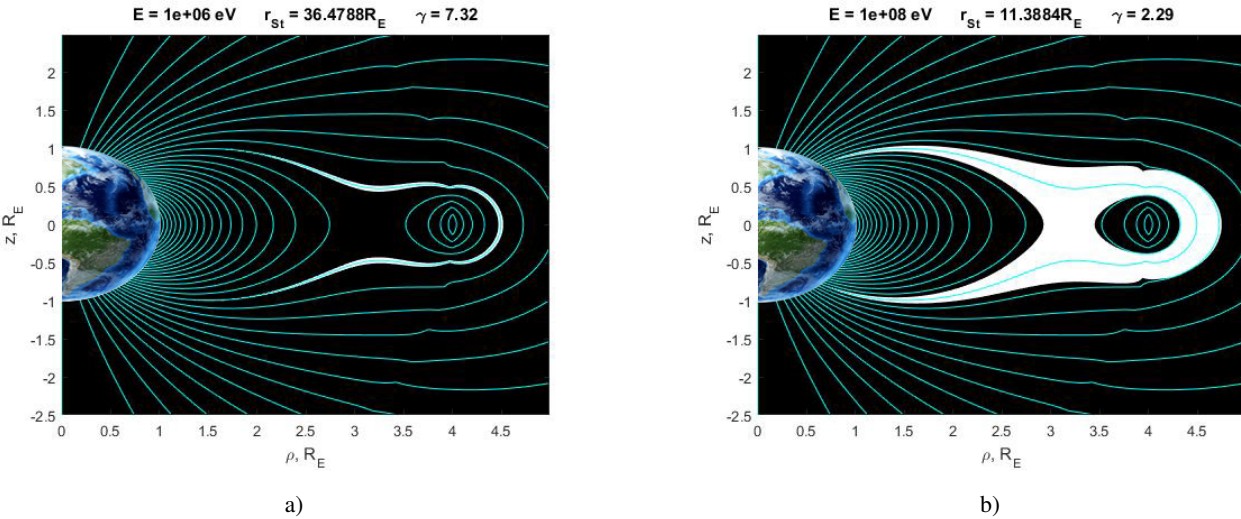

**Figure 7.** The ring current strength $I$ is 10.28 MA, a) 1 MeV, b) 100 MeV

As can be seen from the figures 5-7, the internal trapping regions slightly increase in size with the ring current strength enhancement. Thus particle fluxes of the radiation belts will decrease due to the increase in the spatial dimensions of the allowed region, as observed on Earth during magnetic storms (Kovtyukh and Panasyuk, 2008). At the same time, the depression of the Earth's magnetic field is still not large enough for the particles of the radiation belt to be able to leave the trapping region.

5    This is consistent with the fact that particles can exist in the radiation belts for a long time (Prölss, 2004). We should mention noticeable artifacts in the field lines structure associated with the mathematical expression of the ring current vector potential.

However, the geometry of the allowed-forbidden regions and also the current strength at which the inner and outer Størmer trapping regions get connected strongly depends on two parameters — the external magnetic field $b_z$ and the radius $a$ of the current ring. As an example, we consider the region of motion of the 10 keV proton, in the presence of the ring current of 9.17

10    MA for three magnitudes of the external field $b_z$: -10 nT, -15 nT, and -20 nT (Figures 8a-c):

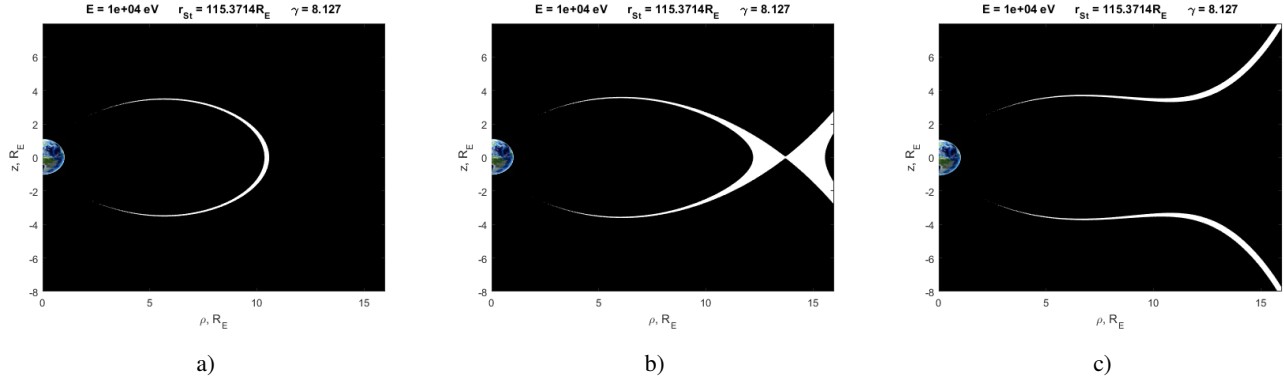

**Figure 8.** The strength of the ring current is 9.17 MA, the energy of the proton is 10 keV. a) $B_z = -10$ nT, b) $B_z = -15$ nT, c) $B_z = -20$ nT





When the field changes by 5 nT, dramatic changes occur - as the $b_z$ decreases, more particles get the opportunity to leave the trapping region much earlier (Fig. 4 c). If, on the other hand, $b_z$ increases, the trapping of particles becomes even stronger - they move closer to Earth, and the outer allowed region of motion moves away from Earth.

Now let us consider how the configuration of the particles' allowed regions of motion varies for different radii $a$ of the

current ring. Again we consider a proton with an energy of 10 keV, and a current strength equal to 9.17 MA; for $a$ we take three different values - $3R_E$, $4R_E$ and $5R_E$ (Figures 9 a-c).

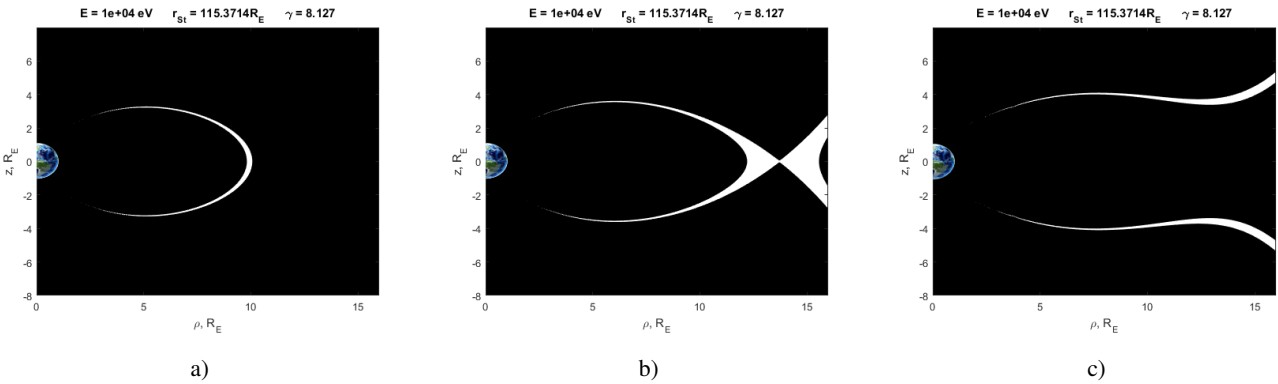

**Figure 9.** The strength of the ring current is 9.17 MA, the particle energy is 10 keV. a) $a = 3R_E$, b) $a = 4R_E$, c) $a = 5R_E$

As the field inside the current ring decreases, and outside increases, we see a logical result. For $a = 4R_E$, the same particles will be in the region of a stronger field (current + dipole) than for $a = 5R_E$ - in this case the field at $L = 8R_E$ is weaker. Therefore, the larger the radius of the model ring with current, the faster the particles at the edge of the real ring current get the

opportunity to leave the trapping region. Conversely, when the radius of the current ring decreases, the field magnitude beyond the distance $a$ becomes larger and the particles are trapped stronger by the magnetic field. Thus, the topology of allowed-forbidden regions and the critical value of Dst index in our problem strongly depends on $b_z$ magnitude and the radius of the ring with the current $a$.

Størmer analysis is not a self-consistent approximation; magnetic field in the problem is axially symmetric and specified —

independent of the trapped particles number. When searching for the critical value of the ring current, we set the distance based on the available observations. Nevertheless, for a complete solution of the problem it would be useful to have a generalization of the Størmer analysis to the case of a self-consistent field. We will try to show, how to come to this solution by the method of successive approximations. We will calculate the threshold current for particles with fixed energy at a given distance, changing the external field. After, we will calculate the threshold current for particles with fixed energy with a given external field,

changing the radius of the ring current $a$. The distance at which the ring current is concentrated will be determined from the actual external field and the observed field at the origin. Having a Dst magnitude, the Dessler-Parker-Sckopke relation (Dessler and Parker, 1959; Sckopke, 1966) can be used to calculate the total ion energy content in the dipole magnetic field, and hence the maximum number of particles that can be trapped by the dipole magnetic field.



Let us consider a proton with energies of 10 keV and 100 keV at $L = 8R_E$. For each $b_z$ magnitude we will search for the current strength magnitude at which the forbidden region between two allowed regions breaks (and the distance at which this link happens):

**Table 1.** Critical strength of a ring current for different magnitudes of the external field $b_z$

| $b_z$, nT | 10 keV proton | | 100 keV proton | |
|---|---|---|---|---|
| | $I_{cr}$, MA | $r_{cr}, R_E$ | $I_{cr}$, MA | $r_{cr}, R_E$ |
| 0 | 1952.7 | 832 | 591.2 | 263.1 |
| -5 | 35.9 | 22.7 | 32.9 | 22.1 |
| -10 | 17.7 | 16.5 | 16.1 | 16.1 |
| -15 | 10.3 | 13.7 | 9.1 | 13.5 |
| -20 | 6.3 | 12.1 | 5.4 | 11.9 |

Let us also consider the influence of the radius $a$ of the ring with current on the critical strength of the current. We again take a proton with energies of 10 keV and 100 keV at $L = 8R_E$ at $b_z = -10$ nT. For each value of the radius of the ring with current, we will search for the current strength magnitude at which the forbidden region between two allowed regions breaks for the protons of the ring current under consideration.

**Table 2.** Critical strength of a ring current for different ring radii $a$

| $a, R_E$ | 10 keV proton | | 100 keV proton | |
|---|---|---|---|---|
| | $I_{cr}$, MA | $r_{cr}, R_E$ | $I_{cr}$, MA | $r_{cr}, R_E$ |
| 3 | 31.8 | 16.4 | 28.9 | 16.1 |
| 4 | 17.7 | 16.5 | 16.1 | 16.1 |
| 5 | 11.2 | 16.5 | 10.2 | 16.2 |
| 6 | 7.6 | 16.6 | 6.9 | 16.2 |
| 7 | 5.5 | 16.6 | 4.9 | 16.3 |

It can be seen from the tables that the field $b_z$ is very important for the determining the critical ring current. The stronger the external field, the faster the ring current breaks and at a lower strength. The radius of the selected ring also has an influence on the strength, for example, for $a = 3R_E$ and $a = 5R_E$, the critical magnitude of the ring current differs by a factor of 3. But



the effect of the external field is much larger. When $b_z$ vanishes, enormous current strength is needed to break the forbidden region. It is also seen from table 1, that with an increase in the negative component of external field, the point where the break occurs is displaced closer to Earth, while the change in the radius of the ring current, as can be seen from Table 2, has almost no effect on the distance to the point of break.

## 6 Conclusions

As a result of particles' injection into the inner magnetosphere, their population, and consequently the ring current strength, increase. Størmer's analysis shows that at a certain point, when the ring current strength reaches critical magnitude, the forbidden region, which separates the internal and external allowed regions disappears, and thus particles get the opportunity to leave the internal trapping region. As a result, the number of particles, which consists the ring current, and consequently the current strength, begin to decrease. In the limit in this case, the particle density near the dipole and at infinity must be the same. The critical magnitude of the ring current's strength depends on the energy of the particles composing the ring current, but at a certain moment the maximum strength is reached, when particles even with the lowest energy are able to leave the trapping region. The maximum possible number of particles that can be trapped in a dipole field is determined by a number of particles, forming a ring current, which changes the "connectivity" of the Størmer regions. For the doubly-connected geometry of the Størmer regions (when internal and external allowed regions of motion are not connected), the particles' flux at infinity and the flux of trapped particles close to Earth can differ arbitrarily. The mixing of passing (coming and going to infinity) and trapped particles does not occur. Therefore, it becomes possible to form regions of trapped radiation, the fluxes of which are determined by the injection and loss mechanisms and are not associated with particles fluxes at infinity. The deformation of the Størmer's regions by the self-consistent field of the trapped radiation (ring current) leads to the formation of connected allowed regions of motion, and in this case the particles' fluxes in the allowed region are uniquely determined by the particles flux and distribution over the pitch angles at infinity. We come to the fact that the ring current strength has a finite upper limit. for At a certain current strength magnitude, the inner trapping region becomes connected to the outer region associated with infinity and further accumulation of particles in the internal trapping region is impossible. At present, there are several recognized main processes, leading to the ring current decay. We showed that a new mechanism, described in the paper, can be also added to these mechanisms. This mechanism leads to the limiting of the ring current strength during a magnetic storm irrespective of the particle loss mechanism from the internal allowed trapping region.

Størmer's analysis shows that the existence of radiation belts follows from the existence of a geomagnetic field. Particles with particle energies of radiation belts and with different parameters $\gamma$ are trapped at different $L-$shells, depending on the Størmer parameter $\gamma$. In addition to the analysis of the critical strength of the ring current, we tried to explain how the Størmer's parameter $\gamma$ is defined and changed. Its change leads to the displacement of the inner trapping region closer to the planet, and, at the same time, to the region's size decrease. The parameter $\gamma$ changes in this case due to the $\boldsymbol{E} \times \boldsymbol{B}$ drift, which moves particles to the inner magnetosphere.



*Competing interests.* The authors declare that they have no conflict of interest.

*Acknowledgements.* Work was supported by the Ministry of Education and Science of the Russian Federation (grant RFMEFI61617X0084)





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
