# Peer review of "Influence of the Earth's ring current strength on the Størmer's allowed and forbidden regions of charged particles motion"

_Annales Geophysicae, 2018_

## Referee Comment (RC1) · Anonymous Referee #1 · 15 Nov 2018

General comments:

This paper contains new approach for evaluation of the influence of the Earth's ring current strength on the Størmer's allowed and forbidden regions based on the solution of the analytical expression, which describes the boundaries of the Størmer's allowed and forbidden regions using the special parameter "gamma". Also, authors discuss how various parameters influence on the geometry of those boundaries and which consequences it has on the particles motion.

- Does the paper contain new data or new ideas or both of them? – new approach.

- Are these up to international standards? – after some corrections.

[Figure]

- Is the presentation clear? – after some corrections.

- Does the author reach substantial conclusions? - yes

- Is the length of the paper adequate? - yes

- Is the language fluent and precise? – after some corrections.

- Are the title and the abstract pertinent and understandable? - yes

- Is the size of each figure adequate to the quantity of data it contains? – yes, except the figure 9.

- Does the author give proper credit to related work and does he/she indicate clearly his/her own contribution? – I would suggest authors to review some of the previous works about the impact of the ring current on the cosmic rays access to the Earth, for instance, the corresponding chapter in the book by Dorman "Cosmic Rays in Magnetosphere of the Earth and other Planets" (2009) and the references included there.

Specific comments:

Page 1, line 17 – Please, correct "one have to..." -> "one has to..."

Page 2, line 18 – "enable" -> "enables"

Page 3, line 20 – here and below I would suggest to rephrase the "south IMF" and "north IMF" to more correct version of this term, for instant, "southward oriented IMF".

Page 3, line 22 – "this phenomena" -> "this phenomenon"

Page 3, line 26 (and the entire chapter 3) – when you mention the index Dst for the first time, please introduce in clearly. The same comment goes to all other variables and parameters used in the paper, especially in the chapter 3: beta, c, m, etc. Some authors prefer to make the list of variables separately in the end of the paper, for example.

Page 3, lines 26, 30, page 4, line 25 – here and below sometimes you write "storm-time ring current", and sometimes "stormtime ring current". Please, use one version.

Page 3, line 27 – please, rephrase the sentence, which contains "After the IMF is rotated to the north…"

Page 4, line 28 – please, rephrase the sentence, which contains "…and the sudden rotation of the IMF to the north…"

Page 5, line 5 – please, remove the parentheses

Page 7, line 19 / p.8 l.20 – "can not" -> "cannot"

Page 8, line 10 – "definition" -> "definition"

Page 9, line 15 – "appears" -> "appear"

Page 16 – Figure 9 is absolutely identical to Figure 8 and does not correspond to the text below and the values in the table 2, probably it is a technical error and another figure should be here?

Page 16, lines 11-13 – here you state that the Dst index strongly depends on bz and the radius of the ring, and on page 18, lines 3-4 you state, that this radius has almost no effect. Please, be consistent and rephrase those sentences and the corresponding text between on pages 16-17.

Page 17 - here you provide two tables, but you don't discuss them in the text. I would suggest to add some comments on those tables.

Page 18, line 21-22 – Please, rephrase the sentence "for At a certain current strength magnitude…"
* * *

---

## Referee Comment (RC2) · Anonymous Referee #2 · 18 Apr 2019

The study by Lavrukhin was very interesting to read. It deals with a generalisation of the Strömer formalism for particle motion in dipole fields by introducing the effects of a variable ring current. It extends its applicability by not only looking into which transient particles from infinity reach into the forbidden regions (and hit the Earth's surface), but also in the trapping and dynamic modulation of the ring current population. In particular, the authors propose a new loss process for ions, developing after the ring current reaches a critical strength. The manuscript has a solid mathematical formulation. Even though my theoretical skills are not that strong, the presentation is clear and to the level I checked, I could not find any mathematical errors. Results are interesting and may readily be applied not just to Earth, but also for other planetary magnetospheres.

Despite that, I do have some reservations about the presentation of the method & theory, having in mind that this paper should be useful not just for researchers with a strong interest in theory, but also to groups dealing with data analysis. Given that, I found it hard to associate certain parameters (e.g. "gamma") with typical observables, such as pitch angle, energy etc. The authors clearly state that particles of same energy can have different gamma, but still that wont be enough for many interested readers. Certain equations describe gamma through the "generalized momentum", to which observers would have trouble to link with observable/measurable quantities e.g. by particle detectrors. Same may apply for the Strömer radius etc. e.g how does it relate to gyroradius. These could either be explained in text, or maybe through an extra illustration/figure.

What I found a bit confusing is the definition of allowed vs. forbidden regions. E.g. in Fig. 1, white shaded areas are allowed for 100 keV protons, but that seems strange at a first sight/read - clearly, inward of the white shaded area, where the field is (much) stronger, trapping is obviously also allowed for 100 keV ions. It may be natural that the trapping limit/limit of adiabaticity may be confused with the character of the allowed region as allowed in this paper. Maybe this ties well to the definition of gamma or energy. E.g. is 100 keV in Fig. 1 the energy at infinity of at the injection point within the magnetosphere? Also, I would be interested to see some comments about the presence of the corotation ExB drift, since the authors do attemp to apply their formulation to keV particles too. Finally, I think a relevant paper may be that of Birmingham 1982 (https://agupubs.onlinelibrary.wiley.com/doi/pdf/10.1029/JA087iA09p07421), where particle motion is analysed in a dipole + ring current, albeit with a different formalism.

Some extra minor comments:

P2, line 5: Change "analytical analysis"

P3, line 19: "directed to the west": "located to the west" is maybe a better expression?

P5 , line 20: do I understand well that the inner boundary of the current is at Earth's
surface?

P9, First lines of section 4: not sure about the explanation of the partial ring current. You write that there is a region where gradient drift dominates, while "in the intermediate region between the gradient and E × B drift regions, a partial ring current arises due to the fact that protons on one side of Earth, and electrons on another have oppositely directed gradient drift velocities.". But the oppositely drift velocities is also a feature of the field gradient, applicable also in the first region where "gradient drift dominates". So, I dont really understand what is the difference between the two regions and where does ExB come in.

P8, line 19: what is it meant "momentum at one Strömer radius?

P9, line 22: add reference in parenthesis

P10 line 13: which value –> the value of which

P18, line 21: correct "for At"

---

## Author Comment (AC3) · 15 May 2019

We would like to post a new, more full answer to the first referee. We have done changes in the paper according to the comments and attach a new pdf file of an article with highlighted changes (in yellow color).

Page 1, line 17 – Please, correct "one have to. . ." -> "one has to. . ."

Thank you, corrected.

Page 2, line 18 – "enable" -> "enables"

Corrected.

[Figure]

Page 3, line 20 – here and below I would suggest to rephrase the "south IMF" and "north IMF" to more correct version of this term, for instant, "southward oriented IMF".

At this point and below the phrases containing the orientation of IMF are corrected.

Page 3, line 22 – "this phenomena" -> "this phenomenon"

Corrected.

Page 3, line 26 (and the entire chapter 3) – when you mention the index Dst for the first time, please introduce in clearly. The same comment goes to all other variables and parameters used in the paper, especially in the chapter 3: beta, c, m, etc. Some authors prefer to make the list of variables separately in the end of the paper, for example.

Dst index is introduced "Dst index (axisymmetric component of the disturbed magnetic field relative to the geomagnetic dipole)", and also other parameters: "Z is the charge of a particle in units of an electron charge, m is the mass of a particle, c - speed of light."

Page 3, lines 26, 30, page 4, line 25 – here and below sometimes you write "storm-time ring current", and sometimes "stormtime ring current". Please, use one version.

Corrected, version storm-time is used.

Page 3, line 27 – please, rephrase the sentence, which contains "After the IMF is rotated to the north. . ."

The sentence is rephrased: "After the IMF changes its direction from southward to the northward, the recovery phase begins. . ."

Page 4, line 28 – please, rephrase the sentence, which contains ". . .and the sudden rotation of the IMF to the north. . ."

The sentence is rephrased: "and the sudden change of the IMF direction to the northward"

Page 5, line 5 – please, remove the parentheses

The reference is changed: "is given by Landau et al. (1982):"

Page 7, line 19 / p.8 l.20 – "can not" -> "cannot"

Corrected.

Page 8, line 10 – "definition" -> "definition"

Corrected.

Page 9, line 15 – "appears" -> "appear"

Corrected.

Page 16 – Figure 9 is absolutely identical to Figure 8 and does not correspond to the text below and the values in the table 2, probably it is a technical error and another figure should be here?

Figure 8 and 9 are very similar, but different. The spatial extent of the allowed region of motion is larger at fig. 8a than at 9a. The difference also exists for fig. 8c and 9c. This is now mentioned in the text after figure 9: "One can see, that figures 9 and 10 are quite similar. Increase of the external field bz leads to the same result as the decrease of the ring current radius a- and to the stronger trapping of particles (Fig. 9a, 10a). Decrease of bz (Fig. 9c) leads to the break of the trapping region, the same happens, when the radius a of the ring current increases (Fig. 9c). As the field inside the current ring decreases, and outside increases, we see a logical result in the Fig. 9a-c. Fora= 4 RE (Fig. 10b), particles will be in the region of a stronger field (ring current field plus dipole) than fora= 5 RE (Fig. 10 c) - in this case the field at L= 8RE is weaker."

Page 16, lines 11-13 – here you state that the Dst index strongly depends on bz and the radius of the ring, and on page 18, lines 3-4 you state, that this radius has almost no effect. Please, be consistent and rephrase those sentences and the corresponding text between on pages 16-17.

The sentences about the effects of the ring current radius and external field on the Dst index are corrected. "Therefore, the larger the radius of the model ring with current, the faster the particles at the edge of the real ring current get the opportunity to leave the trapping region. Conversely, when the radius of the current ring decreases (Fig. 10a), the field magnitude beyond the distance a becomes larger and the particles are trapped stronger by the magnetic field. Thus, the topology of the allowed-forbidden regions of motion and the critical value of Dst index in our problem also strongly depends on the radius a of the ring with the current."

Page 17 - here you provide two tables, but you don't discuss them in the text. I would suggest to add some comments on those tables.

Comments on the Tables 1 and 2 are added to the text.

Page 18, line 21-22 – Please, rephrase the sentence "for At a certain current strength magnitude. . ."

"For" is deleted.

We would like to thank referee again for his opinion about the article and his comments.

Please also note the supplement to this comment:
https://www.ann-geophys-discuss.net/angeo-2018-104/angeo-2018-104-AC3-supplement.pdf

**Supplement:**

[revised manuscript text omitted]

---

## Author Response (AR1)

We would like to thank both referees for their opinion about an article and their comments. We have done changes in the paper according to the comments; at the end of this file there is a new version of a manuscript with highlighted changes (in yellow color).

Answers to the first referee:

*Page 1, line 17 – Please, correct "one have to. . ." -> "one has to. . ."*

Thank you, corrected.

*Page 2, line 18 – "enable" -> "enables"*

Corrected.

*Page 3, line 20 – here and below I would suggest to rephrase the "south IMF" and "north IMF" to more correct version of this term, for instant, "southward oriented IMF".*

At this point and below the phrases containing the orientation of IMF are corrected.

*Page 3, line 22 – "this phenomena" -> "this phenomenon"*

Corrected.

*Page 3, line 26 (and the entire chapter 3) – when you mention the index Dst for the first time, please introduce in clearly. The same comment goes to all other variables and parameters used in the paper, especially in the chapter 3: beta, c, m, etc. Some authors prefer to make the list of variables separately in the end of the paper, for example.*

Dst index is introduced "Dst index (axisymmetric component of the disturbed magnetic field relative to the geomagnetic dipole)", and also other parameters: "Z is the charge of a particle in units of an electron charge, m is the mass of a particle, c - speed of light."

*Page 3, lines 26, 30, page 4, line 25 – here and below sometimes you write "storm-time ring current", and sometimes "stormtime ring current". Please, use one version.*

Corrected, version storm-time is used now.

*Page 3, line 27 – please, rephrase the sentence, which contains "After the IMF is rotated to the north. . ."*

The sentence is rephrased: "After the IMF changes its direction from southward to the northward, the recovery phase begins…"

*Page 4, line 28 – please, rephrase the sentence, which contains ". . .and the sudden rotation of the IMF to the north. . ."*

The sentence is rephrased: "and the sudden change of the IMF direction to the northward"

*Page 5, line 5 – please, remove the parentheses*

The reference is changed: "is given by Landau et al. (1982):"

*Page 7, line 19 / p.8 l.20 – "can not" -> "cannot"*

Corrected.

*Page 8, line 10 – "definition" -> "definition"*

Corrected.

*Page 9, line 15 – "appears" -> "appear"*

Corrected.

*Page 16 – Figure 9 is absolutely identical to Figure 8 and does not correspond to the text below and the values in the table 2, probably it is a technical error and another figure should be here?*

Figure 8 and 9 are very similar, but different. The spatial extent of the allowed region of motion is larger at fig. 8a than at 9a. The difference also exists for fig. 8c and 9c. This is now mentioned in the text after figure 9:

"One can see, that figures 9 and 10 are quite similar. Increase of the external field $b_z$ leads to the same result as the decrease of the ring current radius $a$ - and to the stronger trapping of particles (Fig. 9a, 10a). Decrease of $b_z$ (Fig. 9c) leads to the break of the trapping region, the same happens, when the radius $a$ of the ring current increases (Fig. 9c). As the field inside the current ring decreases, and outside increases, we see a logical result in the Fig. 9a-c. For $a = 4R_E$ (Fig. 10b), particles will be in the region of a stronger field (ring current field plus dipole) than for $a = 5R_E$ (Fig. 10 c) - in this case the field at $L = 8R_E$ is weaker."

*Page 16, lines 11-13 – here you state that the Dst index strongly depends on bz and the radius of the ring, and on page 18, lines 3-4 you state, that this radius has almost no effect. Please, be consistent and rephrase those sentences and the corresponding text between on pages 16-17.*

The sentences about the effects of the ring current radius and external field on the Dst index are corrected.

"Therefore, the larger the radius of the model ring with current, the faster the particles at the edge of the real ring current get the opportunity to leave the trapping region. Conversely, when the radius of the current ring decreases (Fig. 10a), the field magnitude beyond the distance a becomes larger and the particles are trapped stronger by the magnetic field. Thus, the topology of the allowed-forbidden regions of motion and the critical value of Dst index in our problem also strongly depends on the radius a of the ring with the current."

*Page 17 - here you provide two tables, but you don't discuss them in the text. I would suggest to add some comments on those tables.*

Comments on the Tables 1 and 2 are added to the text.

*Page 18, line 21-22 – Please, rephrase the sentence "for At a certain current strength magnitude. . ."*

"for" is deleted.

Answers to the second referee:

*Despite that, I do have some reservations about the presentation of the method & theory, having in mind that this paper should be useful not just for researchers with a strong interest in theory, but also to groups dealing with data analysis. Given that, I found it hard to associate certain parameters (e.g. "gamma") with typical observables, such as pitch angle, energy etc. The authors clearly state that particles of same energy can have different gamma, but still that wont be enough for many interested readers. Certain equations describe gamma through the "generalized momentum", to which observers would have trouble to link with observable/measurable quantities e.g. by particle detectors. Same may apply for the Strömer radius etc. e.g how does it relate to gyroradius. These could either be explained in text, or maybe through an extra illustration/figure.*

We have made a plot of dependence of the Strømer radius of a particle on particle's energy in the Earth's magnetic field. Also, we have considered the gamma parameter properties in more detail in the chapter 4, and showed, how the boundaries of allowed region of motion are determined in Stormer's analysis. Also, we showed at the new plot the value of gamma, which a particle with specified energy will have for at a specified L shell. All these changes are made in chapter 3 and chapter 4 of the paper.

*What I found a bit confusing is the definition of allowed vs. forbidden regions. E.g. in Fig. 1, white shaded areas are allowed for 100 keV protons, but that seems strange at a first sight/read - clearly, inward of the white shaded area, where the field is (much) stronger, trapping is obviously also allowed for 100 keV ions. It may be natural that the trapping limit/limit of adiabaticity may be confused with the character of the allowed region as allowed in this paper. Maybe this ties well to the definition of gamma or energy. E.g. is 100 keV in Fig. 1 the energy at infinity of at the injection point within the magnetosphere?*

In chapter 4 we have shown that the inner allowed region is not the only one and unique for the particle – it can be trapped either closer to the planet, or farther, but with different initial parameters (energy, gamma), which determines its trajectory and allowed region of motion. In order for a particle to get into the currently forbidden region, the parameter gamma must change as a result of some process occurring in the magnetosphere. Thus, the whole allowed region of motion is determined by change of gamma (or generalized momentum) from 0 to +1. The size of the allowed region of motion, which is shown on figures, equals to two the Larmor radius of a particle at a specific point.

*Also, I would be interested to see some comments about the presence of the corotation ExB drift, since the authors do attempt to apply their formulation to keV particles too.*

In the original Størmer analysis magnetic field is stationary and a particle does not receive energy from outside, so the parameter gamma will remain constant. To change it and thus to inject particle to the inner magnetosphere some process is needed. It can be for example either corotation $E \times B$ drift, or the large-scale potential electric fields. These two processes are responsible for the injection of particles to the inner magnetosphere and thus for the change of parameter gamma. Thus, we talk about electric field only in the way that it can be possible reason of change of parameter gamma. We discuss it in the modified chapter 4.

Extra minor comments:

*P2, line 5: Change "analytical analysis"*

Thank you. Accepted, "analytical" removed.

*P3, line 19: "directed to the west": "located to the west" is maybe a better expression?*

Directed to the west describes the direction of the current.

*P5 , line 20: do I understand well that the inner boundary of the current is at Earth's surface?*

To loss the misunderstanding we change the statement to "in the Earth's center". In our approach we propose that ring current localized at distance a from Earth's center. We propose that all current concentrated at ring radius a. Of course, in reality the current distribution localized at some interval of distances but it doesn't change our conclusions.

*P9, First lines of section 4: not sure about the explanation of the partial ring current. You write that there is a region where gradient drift dominates, while "in the intermediate region between the gradient and E x B drift regions, a partial ring current arises due to the fact that protons on one side of Earth, and electrons on another have oppositely directed gradient drift velocities.". But the oppositely drift velocities is also a feature of the field gradient, applicable also in the first region where "gradient drift dominates". So, I dont really understand what is the difference between the two regions and where does ExB come in.*

Accepted. To make the text clearer we corrected this paragraph of Section 4 and didn't consider in detail the processes which leads to the partial ring current formation, because in Stormer's analysis we consider only symmetric ring current.

*P8, line 19: what is it meant "momentum at one Strömer radius?*

Størmer's parameter $\gamma$ is the ratio of the azimuthal components of two different momenta of the particle: $P_\varphi$, taken at infinity, and $pr_s$ at one Størmer radius, with a multiplying factor of 1/2 (taken for the convenience of demonstration); particles of any energy can have the same $\gamma$. In case of the negatively charged particles ($Ze < 0$), the azimuth component of the generalized momentum is directed to the west ($P_\varphi < 0$), and respectively $\gamma < 0$. In case of the positively charged particles ($Ze > 0$), the azimuth component of the generalized momentum is directed to the east ($P_\varphi > 0$), and respectively $\gamma > 0$. In our problem we will consider only protons and therefore gamma values greater than zero.

*P9, line 22: add reference in parenthesis*

Thank you. Accepted.

*P10 line 13: which value –> the value of which*

Accepted: "These two particle'stransport and acceleration mechanisms can act simultaneously Ebihara and Ejiri (2003) and are responsible for the injection ofparticles to the inner magnetosphere and thus for the change of parameter$\gamma$. It will increase when approaching the planet dueto the increase of the magnetic field strength, as mentioned above."

*P18, line 21: correct "for At"*

"for" is deleted.

List of changes

Page 1 line 6:

"*(go to infinity or come to the trapping region from infinity)*" is deleted

Page 1 line 12:

"*(magnetic dipole)*" is inserted

Page 1 line 17:

"*one  has*"

Page 2 line 6:

"*carried out an  analysis*"

Page 2 line 18:

"* enables*"; "*easier access*" is inserted

Page 2 line 26:

"*field*" is inserted

Page 3 line 3:

"*in Chapter 4 we consider the effect of the  Størmer's parameter γ change on the allowed-forbidden regions configuration;*"

Page 3 line 15:

"*the*" is inserted

Page 3 line 20:

"* southward oriented IMF*"

Page 3 line 22:

"* phenomenon*"

Page 3 line 23:

"* southward oriented IMF*"

Page 3 line 26:

Description of Dst index is added "*(axisymmetric component of the disturbed magnetic field relative to the geomagnetic dipole)*"

Page 3 line 28:

"*the IMF  changes its direction from southward to the northward*"

Page 4 line 30:

"*the sudden  change of the IMF direction to the northward*"

Page 4 line 35:

"*(outer, which is connected with the outer space and inner, which isn't)*" is inserted

Page 5 line 8:

" $B = 2\pi I/cR$" formula is converted from SI to CGS.

Page 5 line 9:

" *in the Earth's center*"

Page 6 line 3:

"*m is the mass of a particle, c - speed of light*" is inserted

Page 6 line 14:

"$v^2$" is added

Page 7 line 1:

"*where p is the particle's momentum*" is inserted

Page 7 line 11:

"*condition ... is met*" is inserted

Page 8 line 11:

"$P_{2n+1}^1$ *are the associated Legendre polynomials*" is inserted

Page 8 line 18:

"*In the case of Earth $r_s$ dependence on the particle's energy is shown at Fig. 1. The higher is the energy of the particle, the smaller its Størmer radius.*" is inserted

Page 9 line 0:

Figure is inserted

Page 9 line 1:

" *Størmer's parameter $\gamma$ is the ratio of the azimuthal components of two different momenta of the particle: $P_\phi$, taken at infinity, and*  *the product of momentum $p$ and Størmer radius $r_s$. This ratio is used with a multiplying factor of 1/2 (taken for the convenience of demonstration); particles of any energy can have the same $\gamma$. In case of the negatively charged particles ($Ze < 0$), the azimuth component of the generalized momentum is directed to the west ($P_\varphi < 0$), and respectively $\gamma < 0$. In case of the positively charged particles ($Ze > 0$), the azimuth component of the generalized momentum is directed to the east ($P_\varphi > 0$), and respectively $\gamma > 0$. In our problem we will consider only protons and therefore gamma values greater than zero.*  *having two integrals of motion and the Størmer parameter $\gamma$, one cannot completely describe the trajectory of any particle, but can find the regions of allowed motion and, thus, the trapping regions.*"

Page 9 line 14:

"*With an increase of the particle energy, its Størmer radius $r_s$ decreases.*" is deleted.

Page 10 line 3:

*"Influence of the electric field on particle motion in Størmer's theory: particle injection in the inner magnetosphere"* is changed to *"Størmer's parameter $\gamma$ properties"*

Page 10 line 4 – Page 11 line 10; Page 12 line 1 – Page 12 line 10:

Text and one figure are inserted to clear the meaning of parameter $\gamma$.

Page 13 line 21:

*"Having set the $\gamma$ and energy of the particle (initial parameters), we begin to change the current strength and look what happens with the allowed regions of motion of the particle."* is inserted.

Page 16:

Former figure 6 "The ring current strength I is 9.17 MA, a) 1 MeV, b) 100 MeV" is deleted due to the absence of significant need.

Page 18 line 1:

*"We consider an initial state for a trapped particle under external field $b_z$ = -10 nT (Fig. 9b). When  bz decreases, more particles get the opportunity to leave the trapping region much earlier (Fig. 49 c). If, on the other hand, $b_z$ increases by 5 nT (Fig.9a), the trapping of particles becomes even stronger - they move closer to Earth, and the outer allowed region of motion moves away from Earth. That means, that the magnitude of the external field bz strongly controls the trapping condition for particles in our problem."*

Page 18 line 9-18:

The explanation of figures is expanded and changed.

Page 19 line 9:

*"When $b_z$ vanishes, enormous current strength is needed to break the forbidden region, which will be even larger for a case with external field with positive z component. One can also see that for particles with different energies the point $r_{cr}$, where the break occurs is displaced closer to Earth and gradually converges to a certain value."* was changed to this version.

Page 19 line 18:

*"Although, the effect of the external field on the current strength is much larger, it is obvious, that both parameters $a$ and $b_z$ influences the critical ring current strength, which can exist and, therefore, the Dst index."* is inserted

Page 19 line 21:

*"The change in the radius a of the ring current, as can be seen from Table 2, has almost no effect on the distance $r_{cr}$ to the point, where forbidden region breaks. Thus, having the constant $b_z$ field, there will be a direct dependence between $I_{cr}$ and $a$ (table 2). Now, having dependence between $b_z$ and $I_{cr}$ (from table 1), it becomes possible to calculate the the ring current radius $a$ for a given $b_z$ value."* is inserted

Page 20 line 19:

*" At a certain current"*

Page 21 line 8:

[revised manuscript text omitted]